# Better Objective Sleep Was Associated with Better Subjective Sleep and Physical Activity; Results from an Exploratory Study under Naturalistic Conditions among Persons with Multiple Sclerosis

**DOI:** 10.3390/ijerph17103522

**Published:** 2020-05-18

**Authors:** Dena Sadeghi Bahmani, Roman Gonzenbach, Robert W. Motl, Jens Bansi, Oliver Rothen, Daryl Niedermoser, Markus Gerber, Serge Brand

**Affiliations:** 1Center of Affective, Stress and Sleep Disorders (ZASS), University of Basel, Psychiatric Clinics (UPK), 4002 Basel, Switzerland; daryl.niedermoser@upk.ch (D.N.); serge.brand@upk.ch (S.B.); 2Sleep Disorders Research Center, Kermanshah University of Medical Sciences (KUMS), Kermanshah 6719851115, Iran; 3Departments of Physical Therapy, University of Alabama at Birmingham, Birmingham, AL 35209, USA; robmotl@uab.edu; 4Kliniken Valens, 7317 Valens, Switzerland; Roman.Gonzenbach@kliniken-valens.ch (R.G.); Jens.Bansi@kliniken-valens.ch (J.B.); 5Department of Sport, Exercise, and Health, Division of Sport Science and Psychosocial Health, University of Basel, 4052 Basel, Switzerland; oliver.rothen@gmail.com (O.R.); markus.gerber@unibas.ch (M.G.); 6Substance Abuse Prevention Research Center, Health Institute, Kermanshah University of Medical Sciences (KUMS), Kermanshah 6719851115, Iran; 7School of Medicine, Tehran University of Medical Sciences, Tehran 1416753955, Iran

**Keywords:** sleep-EEG parameters, obstructive sleep apnea, restless legs syndrome, fatigue, naturalistic setting, insomnia

## Abstract

Persons with multiple sclerosis (PwMS) often complain about sleep problems. There is less known about objective sleep-electroencephalography (EEG) dimensions within naturalistic conditions (i.e., home and/or familiar setting). The present cross-sectional study examined the associations between objective and subjective sleep, depression, physical activity scores, and MS-related information among PwMS in their familiar setting. The sample consisted of 16 PwMS (mean age: 50.3 years; median Expanded Disability Status Scale (EDSS): 5.5) who completed questionnaires covering subjective sleep (symptoms of insomnia, restless legs syndrome (RLS) and sleep-disordered breathing), as well as daytime sleepiness, subjective physical activity, depression, and MS-related information (fatigue, EDSS; disease-modifying treatments). Objective sleep was assessed with a mobile sleep-EEG device under naturalist conditions within the home. Descriptively, better objective sleep patterns were associated with lower sleep complaints (r_s_ = −0.51) and daytime sleepiness (r_s_ = −0.43), and with lower symptoms of RLS (r_s_ = −0.35), but not with sleep-disordered breathing (r_s_ = −0.17). More deep sleep was associated with higher moderate physical activity levels (r_s_ = 0.56). Objective sleep parameters were not associated with vigorous physical activity levels (r_s_ < 0.25). Descriptively, moderate and vigorous physical activity scores were associated with lower symptoms of RLS (r_s_ = −0.43 to −0.47). Results from this small study carried out under naturalistic conditions suggest that among PwMS, better objective sleep correlated with better subjective sleep and higher moderate physical activity levels.

## 1. Introduction

Multiple sclerosis (MS) is an immune-mediated, neurodegenerative disease of the central nervous system [1,2]. Persons with MS (PwMS) report a broad range of physical and psychological issues such as depression [3], physical and psychological sleep disturbances [4,5,6], or fatigue [7,8]. For poor sleep, symptoms range from insomnia [5,6,9] and restless legs syndrome [6,10,11] to sleep-related breathing disorders [9,12].

Among non-clinical and clinical samples, poor sleep is associated with a broad range of unfavorable psychological functions such as symptoms of depression [13,14,15,16,17,18] or poor emotion regulation [19]. Among persons with MS (PwMS), poor sleep is associated with daytime sleepiness [20], fatigue [21,22], and cognitive impairments [23,24].

Regarding physical activity, there is sufficient evidence that higher physical activity levels are associated with better objective and subjective sleep, both among non-clinical and clinical samples [14,25,26,27,28,29,30,31], and among PwMS [32,33,34]. Physical activity interventions further have improved both subjective and objective sleep [33,35]. However, when focusing on physical activity and sleep in PwMS, sleep was either assessed subjectively [34], and in only one study, objectively [33].

Another important limitation of previous research is that objective sleep by means of sleep-EEG devices has been assessed in sleep labs, but not under naturalistic conditions, that is, in the natural and/or familiar home setting. Unlike laboratory settings, assessing objective sleep under naturalistic conditions should reflect sleep patterns in a more authentic and reliable manner. Previous studies with clinical samples (borderline personality disorders [36]; patients with ruptured aneurysms post-treatment [37]; children and adolescents with autism spectrum disorder [38]) and non-clinical samples (children [39]; adolescents [40,41]; adolescent children and their parents [42]) showed that the sleep measurements with portable EEG-devices were reliable; furthermore, such devices did not negatively impact on individuals’ subjective sleep quality.

The present study examined objective sleep under naturalistic conditions and examined the association between these objective sleep dimensions with subjective sleep traits and subjective physical activity scores. We tested four hypotheses: (1) We expected that better objective sleep dimensions would correlate with lower self-reported symptoms of insomnia [5,6,9], lower daytime sleepiness [20], lower fatigue [21,22], lower risks of symptoms of restless legs syndrome [6,10,11], and lower sleep-disordered breathing [9,12,43]. (2) We assumed that favorable objective sleep would be associated with lower EDSS scores [33]. (3) We expected that favorable objective sleep dimensions are associated with higher physical activity levels [25,26,30,33,35]. (4) We assumed that higher self-reported physical activity scores would be associated with better subjective sleep [14,25,26,40].

## 2. Methods

### 2.1. Participants and Procedure

Eligible PwMS were fully informed about the aims of the study and about the confidential handling of the data. Thereafter, participants signed the written informed consent. They completed a booklet of self-rating questionnaires covering sociodemographic information, psychological functioning, sleep, and MS-related information (see below for more details). Next, participants’ sleep was assessed with a portable sleep-EEG device at their home, that is, in their natural environment. The ethics committee of Basel and Northwestern Switzerland (EKNZ; Basel, Switzerland) and the ethics committee of St. Gallen (EKOS; St. Gallen, Switzerland) approved the study protocol and its amendments (EKNZ; 2016-011347 a b c). The study was carried out in accordance with the ethical principles laid down in the Declaration of Helsinki and its later amendments (2013). The study was performed between October 2018 and March 2019.

### 2.2. Sample

The sample consisted of PwMS who were formerly treated as inpatients at the Kliniken Valens (Valens, SG, Switzerland) and who took part in an ongoing study on the associations between physical activity, objective and subjective sleep patterns, and psychological functioning among PwMS [33]. Inclusion criteria were: (a) physician-determined diagnosis of MS (following the McDonalds’ diagnostic criteria for MS) [44] irrespective of the subtype (relapse-remitting, primary progressive, and secondary progressive); (b) an age between 18 and 65 years; (c) willing and able to comply with the study requirements as attested by signed written informed consent, including objective sleep assessment with a portable sleep-EEG device; and (d) able to read and write in German. Exclusion criteria were as follows: (a) other neurological or severe psychiatric disorders; and (b) intake of medications apart from disease-modifying treatments and antidepressants.

The severity of the of MS based on EDSS scores was not an exclusion criterion. Furthermore, participants with known musculoskeletal or sleep-disordered breathing were not explicitly excluded as we sought to gain more of an insight into the relationship between physical activity patterns, objective sleep, and sleep-disordered breathing.

Of the 25 eligible PwMS who were approached, 16 of them agreed to participate in the study and provided written informed consent. Nine declined to participate: four indicated time constraints; two planned to undergo further treatments in a rehabilitation center; and three did not provide specific reasons.

### 2.3. Measures

#### 2.3.1. Sociodemographic and MS-Related Information

Participants reported on their age and gender. Furthermore, they reported their current medications, that is: all current disease-modifying medications to treat MS, and psychopharmacological medications to treat psychiatric symptoms such as depression, sleep disturbances, or chronic pain. Next, participants were asked to list current MS-related treatments such as physiotherapy, osteopathy, acupuncture, exercising, massage, dieting, or others. Lastly, participants were instructed to ask their neurologists about their current EDSS score.

#### 2.3.2. Objective Sleep

As in previous studies [33,42], sleep was objectively assessed with a one-channel, portable sleep-EEG recording device (Fp2-A1; electromyogram; electrooculogram; Somnowatch^®^; Somnomedics; Randersacker, Germany). These simple though powerful devices have been shown to provide reliable data [36]; two experienced sleep lab experts visually analyzed the EEG signals according to standard procedures [45]. Sleep continuity parameters consisted of: (a) sleep onset latency (min); (b) total sleep time (min); and (c) duration of awakenings after sleep onset (min). Sleep architecture parameters were: (a) non-rapid eye movement sleep (non-REM sleep); (b) light sleep (stages 1 and 2): absolute (min) and relative duration (%); (c) slow wave sleep/deep sleep (stages 3 and 4): absolute (min) and relative duration (%); and (d) rapid eye movement sleep (REM): absolute (min) and relative duration (%). All measurements were reported in means (M) and standard deviations (SD).

#### 2.3.3. Subjective Sleep

##### Symptoms of Insomnia

Participants completed the Insomnia Severity Index (ISI) [46,47]. This questionnaire consists of seven items and refers in part to the Diagnostic and Statistical Manual of Mental Disorders [48] to define insomnia. Items ask about difficulty falling asleep, difficulty remaining asleep, early morning awakenings, impaired daytime performance, low satisfaction with sleep, and worry about sleep. Answers are given on a five-point Likert scale ranging from 0 (= not at all) to 4 (= very much), with higher overall scores reflecting a higher insomnia severity (M; SD). The Cronbach’s α for the present sample was 0.84.

##### Daytime Sleepiness

To assess daytime sleepiness, participants completed the Epworth Sleepiness Scale (ESS) [49] which consists of ten items rating the odds of dozing off in different activities. Answers are given on 4-point rating scales (0–3), with higher sum scores reflecting higher daytime sleepiness. The sum score ranges were from 0 to 30 points; a global score greater than 10 indicated excessive daytime sleepiness. ESS scores were treated as continuous. The validity and reliability of this instrument have been established previously [50] (M; SD). The Cronbach’s α for the present sample was 0.83.

##### Restless Legs Syndrome

The International Restless Legs Study Group (IRLSSG) rating scale was administered in a face-to-face interview following the procedure as set out by the IRLSSG [51]. Questions focus on RLS-related severity such as RLS discomfort in legs and arms, the need to move around for relief from RLS, and the frequency of RLS. Answers are given on 5-point Likert scales ranging from 0 (none) through to 4 (very severe), with higher scores reflecting more marked symptoms of RLS (M; SD). In the present sample, the internal consistency of the instrument was satisfactory (Cronbach’s α = 0.80).

##### Self-Assessed Risks of Snoring and Sleep Apnea

The STOP-Bang questionnaire [52] is a self-rating tool to assess the risk of snoring and sleep apnea. Answers are given on forced-choice (yes/no) scales to rate the risks of suffering from obstructive sleep apnea. The questionnaire is composed of four questions related to snoring (S), tiredness during the daytime (T), observed apneas (O), and high blood pressure (P). When two or more questions were answered with “yes”, that person had an increased risk of obstructive sleep apnea/hypopnea syndrome. The “Bang” portion is evaluated by assessing BMI > 35 kg/m^2^ (B), age (>50 years) (A), neck circumference (>40 cm) (N), and gender (male) (G). One point is assigned for each positive answer and zero for each negative answer. High risk for the Obstructive Sleep Apnea Syndrome (OSAS) on the STOP-Bang questionnaire is indicated when three or more of the eight questions are answered with “yes”. In the present study, we reported the total score (M; SD). The STOP-Bang questionnaire appears to be a reliable screening tool for OSAS [53].

#### 2.3.4. Psychological Functioning

##### Depressive Symptoms

As previously used with PwMS [33], participants completed the Beck Depression Inventory–Fast Screen (BDI-FS) [54] to assess symptoms of depression. The BDI-FS consists of seven items, and every item has a set of four possible responses, representing different levels of symptom severity (e.g., sadness: 0 = “I don’t feel sad”; 1 = “I feel sad”; 2 = “I’m sad all the time and I can’t snap out of it”; 3 = “I’m so sad/unhappy, that I can’t stand it”). Higher scores reflect a greater severity of depressive symptoms (range: 0–21; M; SD). The validity and reliability of this instrument have been established in prior studies [55]. The Cronbach’s α for the present sample was 0.80.

##### Fatigue

As in a previous study [33], participants completed the Fatigue Severity Scale (FSS) [56]. This consists of nine items, and answers are given on seven-point rating scales ranging from 1 (= not at all) to 7 (= definitively/almost always), with higher scores reflecting higher levels of fatigue (M; SD). Psychometric properties of the FSS are satisfactory [57]. In the present sample, the Cronbach’s α was 0.98.

#### 2.3.5. Physical Activity

The short form of the International Physical Activity Questionnaire (IPAQ-SF) is a self-rating questionnaire used to assess subjective physical activity [58]. As outlined elsewhere [59], participants reported on how many days per week they performed vigorous physical activity; “vigorous” was operationalized with: “After vigorous physical activity, I’m sweating, breathing a lot, and I feel tired”. The response categories ranged from 0 to 7 days. In addition, participants were asked to indicate the average duration per day of their exercise on the days they engaged in these activities. Multiplication of frequency and duration scores resulted in an estimate of weekly hours invested in vigorous physical activity. To assess moderate physical activity, the items and the calculations were identical, though “moderate” was operationalized with: “After moderate physical activity, I feel a bit tired, though, I can breathe normally, and I do not feel tired.” Scores are always reported in M and SD.

### 2.4. Statistical Analysis

All statistical computations were performed with SPSS^®^ 25.0 (IBM Corporation, Armonk NY, USA) for Apple Mac^®^. A series of Spearman’s correlations was performed between objective sleep dimensions and subjective sleep dimensions, including daytime sleepiness and fatigue, and along with EDSS scores and physical activity scores. The nominal level of significance was set at alpha < 0.05.

## 3. Results

### 3.1. Sociodemographic and MS-Related Information

Table 1 reports the descriptive statistical indices of sociodemographic and MS-related information.

Participants’ mean age was 50.25 years. Eleven (69%) out of 16 participants were females. The median EDSS score was 5.25. Participants used both disease-modifying and non-disease modifying medications. Ten (62.5%) out of 16 participants did not take psychopharmaceuticals.

### 3.2. Correlations between Objective Sleep and Subjective Sleep, Psychological Dimensions, and Physical Activity Levels

Table 2 provides the descriptive statistical indices and the Spearman’s rho correlation coefficients (r_s_) between objective sleep dimensions and subjective sleep dimensions, self-reported symptoms of restless legs syndrome, self-reported sleep-disordered breathing, daytime sleepiness, and fatigue. Note that the Spearman’s rho correlation coefficients (r_s_) are not repeated in the text again.

Descriptively, a longer total sleep time was associated with lower daytime sleepiness, lower RLS scores, lower symptoms of depression, and higher moderate physical activity levels. Trivial-to-small correlational coefficients were observed for age, BMI, subjective sleep complaints, sleep-disordered breathing, fatigue, and vigorous physical activity.

A higher sleep efficiency was associated with lower subjective sleep complaints (always *p* < 0.05). Descriptively, a higher sleep efficiency was associated with lower daytime sleepiness, lower RLS scores, and higher vigorous PA levels. Trivial-to-small correlational coefficients were observed for age, BMI, sleep-disordered breathing, depression, fatigue, and moderate physical activity.

Descriptively, a shorter sleep onset latency was associated with lower subjective sleep complaints, a lower daytime sleepiness, and lower RLS scores. Trivial-to-small correlation coefficients were observed for age, BMI, sleep-disordered breathing, depression, fatigue, and moderate and vigorous physical activity levels.

A longer wakening time after sleep onset was associated with lower moderate physical activity levels. For all other dimensions, correlation coefficients were trivial to small.

A longer REM-sleep time was associated with a higher age, and lower vigorous physical activity levels. For all other dimensions, correlation coefficients were trivial to small. A higher relative amount of REM-sleep (%) was associated with higher sleep complaints, higher symptoms of depression, and a higher daytime sleepiness. For all other dimensions, correlation coefficients were trivial to small.

A longer light sleep was associated with higher daytime sleepiness, and higher moderate physical activity levels. For all other dimensions, correlation coefficients were trivial to small. A higher amount of light sleep (%) was associated with a lower daytime sleepiness, lower symptoms of depression and higher moderate physical activity levels. For all other dimensions, correlation coefficients were trivial to small.

A longer deep sleep was associated with higher moderate physical activity levels. For all other dimensions, correlation coefficients were trivial to small. A higher amount of deep sleep (%) was associated with lower daytime sleepiness and lower moderate physical activity levels. For all other dimensions, correlation coefficients were trivial to small.

To summarize, dimensions of better objective sleep were associated with lower subjective sleep complaints, less daytime sleepiness, lower self-reported symptoms of RLS, and higher moderate physical activity levels.

### 3.3. Correlations between Objective Sleep and EDSS Scores

Table 2 further indicates that higher EDSS scores were associated with a higher sleep efficiency, a shorter sleep onset latency, and with longer wake time after sleep onset. 

### 3.4. Correlations between Self-Reported Physical Activity Levels and EDSS Scores, Subjective Sleep-Related Dimensions and Psychological Dimensions

Descriptively, higher moderate physical activity levels were associated with lower sleep complaints, and lower symptoms of RLS and fatigue.

Descriptively, higher vigorous physical activity levels were associated with lower EDSS scores, lower sleep complaints, and lower symptoms of RLS, and higher daytime sleepiness.

## 4. Discussion

The key findings of the present cross-sectional study were that favorable objective sleep parameters were associated with dimensions of favorable subjective sleep (lower sleep complaints, lower daytime sleepiness, and lower self-reported symptoms of RLS) among a small sample of PwMS. Better objective sleep correlated with higher self-reported physical activity levels. Against expectations, better objective sleep was associated with higher EDSS scores. The present data add to the current literature in an important way, as we were able to show that objective sleep assessed under naturalistic conditions was associated with subjective sleep dimensions, along with self-reported physical activity levels.

Regarding the first hypothesis, we assumed that better objective sleep would be associated with a broad range of favorable subjective sleep dimensions, and our data supported this assumption. The present data are further in line with previous results on lower self-reported symptoms of insomnia [5,6,9], lower daytime sleepiness [20], and lower risks of symptoms of restless legs syndrome [6,10,11]. However, against findings reported elsewhere [9,12], better objective sleep was unrelated to self-reported scores of sleep-disordered breathing and fatigue [21,22]. As regards the latter result, in the absence of direct evidence of the underlying psychophysiological mechanisms, we offer the following admittedly speculative interpretations. First, as shown in Table 2, prevalence rates, scores, and the range of self-reported sleep-disordered breathing were low; accordingly, it seems plausible that possible floor effects and low variance precluded meaningful correlations. Second, OSAS were based on self-reports, but an objective assessment of OSAS would have allowed us to rely on robust data. Likewise, validity and reliability of the German version of the STOP-Bang questionnaires are moderately satisfactory [53], thus, bearing the risk of not appropriately reflecting symptoms of sleep-disordered breathing.

Regarding the second hypothesis, we assumed that better objective sleep would be associated with lower EDSS scores; however, data supported the opposite association: better objective sleep was associated with higher, but not lower, EDSS scores. This result was not as expected, as reviews and meta-analyses [6,9] draw the conclusion that higher EDSS scores are associated with more impaired sleep.

In contrast, and as regards subjective sleep dimensions listed in Table 3, this assumption could be partially confirmed: higher EDSS scores were associated with higher sleep-disordered breathing, higher symptoms of RLS, and lower vigorous physical activity scores. However, and again, the quality of the data does not allow a deeper understanding of the unexpected result. Speculatively, it is possible that EDSS scores did not reflect participants’ current state of disability.

Regarding the third hypothesis, we assumed that better objective sleep should be associated with higher physical activity scores, and data did confirm this. Accordingly, the present results are in accord with studies on non-clinical samples of adolescents [30,38,40,60] and adults [25,26,31]. The present data expands upon previous results in that such patterns were observed among a smaller sample of PwMS assessed in their natural in-home conditions.

Regarding the fourth hypothesis, we assumed that subjective sleep dimensions would be associated with physical activity scores, and the data confirmed this assumption. It follows that the present pattern of results is in accord with the host of previous studies with non-clinical and psychiatric samples, and with samples of PwMS [25,26,30,33,35]. However, the present data expands upon previous results in that objective sleep dimensions were not assessed in a sleep lab, but in participants’ familiar environment at home. In this view, Bruyneel and Ninane [61] concluded in their review that objectively assessed sleep at home tended to be better, compared to objectively assessed sleep assessed in a sleep lab. Likewise, we hold that the present data should be more reliable compared to sleep data assessed in participants’ unusual and artificial environment of a sleep lab [62].

The novelty of the results should be balanced against the following limitations: first, the sample size was small; it follows that it relied on the directions of the correlation coefficients; with a larger sample size the correlation coefficients would have remained unaltered, though statistical significance might have been reached [63]. Second, as mentioned, it is conceivable that the EDSS scores did not reflect the current state of disability. Likewise, third, symptoms of RLS and sleep-disordered breathing were subjectively assessed; it follows that such subjective ratings might rather reflect risks of RLS and sleep-disordered breathing disorders, but not an accurate diagnosis, as ascertained with objective criteria in a sleep-lab and by trained neurologists, pneumologists, and clinical psychologists. It further follows that in future, studies on symptoms of RLS and sleep-disordered breathing should be thoroughly diagnosed based on objective assessments. Regarding OSAS, the gold standard is the apnea/hypopnea index (AHI; cf. [64]), which, however, needs to be thoroughly assessed in a sleep lab with polysomnography and was beyond the scope of our naturalistic, home-based study. Furthermore, as regards the German version of the STOP-Bang, psychometric properties were moderately satisfactory [53]. Fourth, other latent, but unassessed physiological, psychological (e.g., motivation; curiosity), or MS-related dimensions (age at disease onset; disease duration; number of relapses; current state of MS: remitting-relapsing MS; primary progressive MS; secondary progressive MS; number of attacks; time lapse to last attack) might have biased two or more dimensions in the same or opposite directions. Last, the cross-sectional character of the study precludes us from drawing causal effects from it.

We hold that the present results are clinically important, because besides the study of Veauthier et al. [22], it appears to be only the second study to assess objective sleep of PwMS under naturalistic settings, and the first study to assess objective sleep under naturalistic conditions in relation to PwMS’s physical activity patterns. This means that the present results give a closer view to objective sleep patterns of PwMS. The present results are also of practical importance, given that regular physical activity has the potential to improve sleep also among PwMS [33].

## 5. Conclusions

Among a smaller sample of PwMS, objective sleep parameters assessed under naturalistic conditions were associated with better subjective sleep, and higher levels of self-reported physical activity.

## Figures and Tables

**Table 1 ijerph-17-03522-t001:** Descriptive statistical indices of sociodemographic and MS-related information.

Variables	
	M (SD)
Age (years)	50.3 (9.6)
BMI (kg/m^2^)	28.6 (5.4)
	*n* (%)
Gender (female/male)	11/5 (68.8/31.2)
	Median (range)
EDSS	5.5 (3–7)
Medications	
	*n* (%)
No disease-modifying treatments such as tizanidine (and natalizumab)	8 (50%)
Disease-modifying treatments such as interferon beta, glatiramer acetate, dimethyl fumarate, teriflunomide fingolimod, (natalizumab), rituximab, ocrelizumab, and mitoxantrone	8 (50%)
Psychopharmaceuticals, including pain killers and phytopharmaceuticals	
No psychopharmaceuticals	10 (62.5)
Phytopharmaceuticals	3 (18.8)
Pregabalin (Lyrica^®^)	1 (6.3)
Lamotrigine (Lamictal^®^)	2 (12.5)
OTC pain killers	6 (37.5)

Notes: BMI = Body Mass Index; EDSS = Expanded Disability Status Scale; OTC = over the counter.

**Table 2 ijerph-17-03522-t002:** Spearman’s correlation coefficients between objective sleep parameters (sleep-EEG data) and age, EDSS, BMI, sleep complaints, daytime sleepiness, symptoms of restless legs syndrome (RLS) and sleep-disordered breathing (SDB), depressive symptoms, fatigue, and self-reported physical activity levels.

					Dimensions							Statistics
	Age	EDSS	BMI	ISI	ESS	RLS	SDB	BDI-FS	Fatigue	PA Moderate	PA Vigorous	M (SD)
Total sleep time (min)	−0.07	−0.03	0.13	0.19	−0.25	−0.30	0.10	−0.26	−0.03	0.49	0.01	417.44 (62.68)
Sleep efficiency (%)	0.16	0.52 *	−0.05	−0.51 *	−0.43	−0.35	−0.17	0.13	0.28	0.01	0.34	81.06 (9.67)
Sleep onset latency (min)	0.02	−0.25	−0.02	0.40	0.44	0.49	0.17	−0.11	0.22	−0.30	−0.26	15.51 (10.40)
WASO time (min)	0.14	0.33	−0.12	0.14	0.16	0.09	0.12	0.28	−0.19	−0.27	0.18	42.24 (4.03)
REM sleep time (min)	0.30	−0.17	0.16	−0.13	−0.13	−0.14	0.19	0.10	−0.09	0.20	−0.43	23.07 (4.02)
REM (%)	0.29	0.33	0.23	0.40	0.41	−0.42	−0.00	0.39	−0.07	0.27	0.04	5.13 (0.99)
Light sleep duration (min)	−0.27	−0.12	−0.21	−0.02	0.37	0.05	−0.04	0.42	−0.14	0.40	0.24	276.13 (45.50)
Light sleep percentage (%)	−0.32	−0.12	−0.09	0.07	−0.35	−0.03	−0.06	−0.38	−0.23	0.51 *	−0.16	58.00 (5.65)
Deep sleep duration (min)	0.32	−0.16	−0.07	−0.28	0.18	0.07	0.22	0.02	0.03	0.56 *	0.25	87.97 (23.26)
Deep sleep percentage (%)	0.43	0.16	−0.02	−0.04	−0.43	0.05	0.05	0.23	0.07	−0.71 **	−0.03	19.43 (4.86)

Notes: EDSS = Expanded Disability Status Scale; BMI = Body Mass Index; BDI-FS = Beck Depression Inventory; Fast Screen; ISI = Insomnia Severity Index; ESS = Epworth Sleepiness Scale; RLS = Restless Legs Syndrome; SDB = Sleep-disordered breathing; PA = physical activity. REM = rapid eye movement; WASO = wakening after sleep onset. * = *p* < 0.05; ** = *p* < 0.01. Always Spearman’s rho correlation coefficients.

**Table 3 ijerph-17-03522-t003:** Descriptive statistical indices and Spearman’s correlation coefficients of age, EDSS, BMI, sleep complaints (ISI), daytime sleepiness (ESS), symptoms of restless legs syndrome (RLS), and sleep-disordered breathing (SDB), depressive symptoms (BDI), fatigue, and self-reported physical activity levels.

Dimensions												Statistics
	Age	EDSS	BMI	ISI	ESS	RLS	SDB	BDI	Fatigue	PA Moderate	PA Vigorous	M (SD)
Age (years)	-	0.23	0.18	0.14	0.20	−0.09	0.27	0.09	−0.16	−0.06	0.27	50.25 (9.59)
EDSS		-	−0.34	0.50 *	0.54 *	0.42	0.44	0.22	0.41	−0.06	−0.61 *	5.28 (1.19)
BMI			-	0.18	0.02	0.18	0.83 **	0.35	0.08	0.26	−0.10	29.59 (5.40)
ISI				-	0.69 **	−0.38	0.16	0.38	0.36	−0.46	−0.66 *	15.12 (6.48)
ESS					-	−0.27	0.05	0.71 *	0.45	−0.03	0.49	8.37 (5.19)
RLS						-	0.28	0.15	0.20	−0.47	−0.43	15.81 (1.32)
SDB							-	0.37	0.02	0.13	−0.14	1.81 (0.75)
BDI									0.53 *	−0.08	−0.02	9.44 (4.89)
Fatigue									-	−0.30	0.09	48.63 (16.26)
PA moderate (min)										-	0.35	328.25 (369.40)
PA vigorous (min)											-	105.94 (106.15)

Notes: EDSS = Expanded Disability Status Scale; BMI = Body Mass Index; BDI = Beck Depression Inventory Fast Screen; ISI = Insomnia Severity Index; ESS = Epworth Sleepiness Scale; RLS = Restless Legs Syndrome; SDB = Sleep-disordered breathing; PA = physical activity. * = *p* < 0.05 ** = *p* < 0.01.

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
