# Peer review of "Better Objective Sleep Was Associated with Better Subjective Sleep and Physical Activity; Results from an Exploratory Study under Naturalistic Conditions among Persons with Multiple Sclerosis"

_ijerph, 2020, doi:10.3390/ijerph17103522_

Round 1

Reviewer 1 Report

I think that the study entitled "Better objective sleep was associated with better subjective sleep and physical activity; results from an exploratory study under naturalistic conditions among persons with multiple sclerosis" provides interesting results, but need to improve some aspects for their acceptation to publish in IJERPH.

Major comments

  1. Please, add a flowchart with the recruitment process
  2. The authors excluded patients with other neurological disorders but, were included patients with musculoskeletal disorders? Because it affects physical activity. Or were excluded patients with respiratory diseases? Because it affects physical activity.
  3. Are there patients with mobility restrictions? For example, in a wheelchair?
  4. In statistical analysis, it is necessary to describe how the authors present continuous or dichotomic variables, mean, and SD? Measures of frequency? Median? IQR?
  5. Please, present all data with only one decimal. For example, not 50.25 years, only 50.3 years
  6. How long were the patients diagnosed?
  7. In table 1 said that males are 11, but in line 204 say that females are 11. Please review the data.
  8. The most crucial variable in sleep is the Apnea-hypopnea index. For why reason the authors do not use AHI?
  9. In table 3, the authors present the correlation results ant the descriptive statistics. Please separate in different tables since they are confusing to the reader.
  10. Eighty references are excessive for this manuscript. Please reduce to only necessary references.

Minor comments

  1. Please describe the abbreviation EDSS the first time that appears
  2. Line 131. Methods. Replace (Gerber et al., 2016) by [number]
  3. In gender distribution say 68.8% of male and 31.3% of female….the sum is 100.1%
  4. In table 1, add the abbreviations (EDSS, BMI)
  5. Line 295. Review the reference, and replace by [number]

Author Response

We thank Reviewer #1 for her/his valuable comments.We have address all issues;  please see the detailed point-by-point-response attached as a separate file. 

Reviewer 2 Report

Thank you for the opportunity to review this manuscript. Overall, this is an important topic where the objective was to examine the associations between objective and subjective sleep, depression, physical activity scores and MS-related information among persons with multiple sclerosis in their familiar setting.

I don't feel qualified to judge about the English language and style. There is good data in this study. The article is very interesting and mainly well-written. There are only some notes in my assessment. There are some minor suggestions to improve the manuscript:

First, In the Abstract section, please consider introduce some additional information about the results (Spearman’s correlations  and significance or “p” of the main results).

Many abbreviations have been used in the abstract that have not been previously defined. Some examples: EDSS, RLS… Abbreviation should be the first time these terms are used in the text.

The conclusions in the abstract are repetitive. Maybe you should delete the last sentence

 “Results add to the sparse literature on objective sleep dimensions carried out under naturalistic conditions”.

  1. Introduction

The introduction provide sufficient background. It has included relevant and new references (last years).

It has provided a conceptual framework to help the reader understand the manuscript and the objectives. They have explained this information has been reviewed in others studies.

The hypotheses are adequately described. The research design is appropriate.

  1. Methods

Describing the Study design. Please provide a date at which study enrolment commenced.
When did you interview/survey the patients with the instruments or questionnaires?

  1. Results

The presentation of Spearman’s correlations and significance or “p” of the main results described in the text, would improve the interpretation of the results or remark this principal information in the tables.

  1. Discussion
    In the discussion section on line 299, you forgot to enter a reference. It is marked.

The conclusions are supported by the results.

Author Response

We thank Reviewer #2 for her/his valuable comments.We have address all issues;  please see the detailed point-by-point-response attached as a separate file. 

Round 2

Reviewer 1 Report

The authors have substantially improved the quality of the manuscript. I suggest its publication.